# Exosomes in Vascular/Neurological Disorders and the Road Ahead

**DOI:** 10.3390/cells13080670

**Published:** 2024-04-12

**Authors:** Faisal A. Alzahrani, Yasir M. Riza, Thamir M. Eid, Reema Almotairi, Lea Scherschinski, Jessica Contreras, Muhammed Nadeem, Sylvia E. Perez, Sudhanshu P. Raikwar, Ruchira M. Jha, Mark C. Preul, Andrew F. Ducruet, Michael T. Lawton, Kanchan Bhatia, Naseem Akhter, Saif Ahmad

**Affiliations:** 1Department of Biochemistry, King Fahad Center for Medical Research, King Abdulaziz University, Jeddah 21589, Saudi Arabia; 2Department of Medical Laboratory Technology, Prince Fahad bin Sultan Chair for Biomedical Research, Faculty of Applied Medical Sciences, University of Tabuk, Tabuk 71491, Saudi Arabia; 3Department of Translational Neuroscience, Barrow Neurological Institute, St Joseph’s Hospital and Medical Center, Phoenix, AZ 85013, USAjessica.contreras@barrowneuro.org (J.C.);; 4Department of Neurology, Barrow Neurological Institute, St Joseph’s Hospital and Medical Center, Phoenix, AZ 85013, USA; 5Department of Neurosurgery, Barrow Neurological Institute, St Joseph’s Hospital and Medical Center, Phoenix, AZ 85013, USA; 6School of Mathematical and Natural Sciences, Arizona State University, Glendale, AZ 85306, USA; 7Department of Biology, Arizona State University, Lake Havasu City, AZ 86403, USA; 8Phoenix Veterans Affairs (VA) Health Care System, Phoenix, AZ 85012, USA

**Keywords:** exosomes, vascular diseases, neurological diseases, stroke, Alzheimer’s

## Abstract

Neurodegenerative diseases, such as Alzheimer’s disease (AD), Parkinson’s disease (PD), amyotrophic lateral sclerosis (ALS), Huntington’s disease (HD), stroke, and aneurysms, are characterized by the abnormal accumulation and aggregation of disease-causing proteins in the brain and spinal cord. Recent research suggests that proteins linked to these conditions can be secreted and transferred among cells using exosomes. The transmission of abnormal protein buildup and the gradual degeneration in the brains of impacted individuals might be supported by these exosomes. Furthermore, it has been reported that neuroprotective functions can also be attributed to exosomes in neurodegenerative diseases. The potential neuroprotective functions may play a role in preventing the formation of aggregates and abnormal accumulation of proteins associated with the disease. The present review summarizes the roles of exosomes in neurodegenerative diseases as well as elucidating their therapeutic potential in AD, PD, ALS, HD, stroke, and aneurysms. By elucidating these two aspects of exosomes, valuable insights into potential therapeutic targets for treating neurodegenerative diseases may be provided.

## 1. Extracellular Vesicles and Exosome Biogenesis

Extracellular vesicles (EVs) are generally divided into two main types: ectosomes and exosomes. Ectosomes result from the direct outward budding of the plasma membrane, yielding larger vesicles sized around 50 nm to 1 μm. In contrast, exosomes, originating from endosomes, are smaller, typically ranging from 40 to 160 nm in diameter, with an average size of 100 nm. The focus of the current review is on exosomes and their roles in neurological disease. Exosomes form through the double invagination of the plasma membrane and the creation of intracellular multivesicular bodies (MVBs) containing intraluminal vesicles (ILVs). These ILVs are then released as exosomes within the size range of approximately 40 to 160 nm in diameter. The biogenesis of exosomes includes the maturation of early-sorting endosomes (ESEs) into late-sorting endosomes (LSEs), ultimately leading to MVB formation [1,2,3]. MVBs can fuse with lysosomes or autophagosomes for degradation or with the plasma membrane to release ILVs as exosomes (Figure 1) [4]. Several proteins and molecules, including Rab GTPases, Syntenin-1, TSG101, ALIX, syndecan-1, ESCRT proteins, phospholipids, tetraspanins, ceramides, sphingomyelinases, and SNARE complex proteins, are involved in the biogenesis and origin of exosomes [3,5,6]. The complexity of studying exosome biogenesis arises from its interaction with various molecular pathways in intracellular vesicle trafficking. Experiments altering regulatory elements in exosome biogenesis may indirectly impact other cellular processes, including autophagy, lysosomal pathways, and Golgi apparatus-derived vesicle trafficking [3]. Differences in cell types, culture conditions, and genomic health can impact exosome biogenesis regulation. Quantifying exosome production rates is challenging due to the dynamic nature of both the de novo production and cell uptake of external exosomes. Studies have shown varying rates of net exosome production by different cell types, with cancer cells generally secreting more exosomes compared to normal cells [5].

## 2. Cellular Processes and Intercellular Communication

Exosomes play a crucial role in understanding their cargo’s fate and the molecular changes they induce in recipient cells. Uptake and secretion pathways can overlap, creating a diverse population of both endogenously produced and recycled exosomes. Mechanisms for exosome uptake by recipient cells, such as macropinocytosis, fusion with the plasma membrane, and clathrin-dependent endocytosis, can result in varied outcomes for exosomal cargo [7,8]. The specificity of exosomes for certain cell types adds complexity to their function in intercellular communication. In vivo experiments using genetically modified mice have revealed rare instances of exosomes delivering mRNA to recipient cells, particularly in the context of inflammation and tumor growth [9]. The protein cargo of exosomes can be altered in cancer cells, leading to the enrichment of specific proinvasive molecules or signaling molecules that activate specific pathways in recipient cells (Figure 2) [10]. These findings highlight the selective sorting of proteins into exosomes and their potential to regulate various biological processes in development, immune responses, and disease [11].

## 3. Role of Exosomes in Various Biological Processes

### 3.1. Cell Autonomous Processes

Exosome biogenesis enables cells to swiftly and selectively remove proteins from the plasma membrane. Instances include the fertilization-triggered release of the sperm receptor Juno and the controlled elimination of transferrin receptor (TfR) during reticulocyte plasma membrane processes [19,20,21]. Exosomes also play a role in protein quality control, facilitating the release of the cell death mediator MLKL, expelling unshielded and chemically modified RNAs, and vesicular secretion of neurodegenerative amyloidogenic proteins and aggregated signaling molecules [22,23]. Exosomes also play a role in establishing anterior–posterior cell polarity, particularly in highly migratory cells. This type of cell polarity, which occurs rapidly in response to polarizing stimuli, involves the exosome biogenesis pathway. This pathway selectively sorts and enriches exosomal proteins and lipids at the uropod (posterior pole) of amoeboid cells. This polarized sorting facilitates rapid and directional cell migration, aiding cells like leukocytes and amoeboid cells in quickly establishing morphological and functional polarity [24,25].

### 3.2. Remodeling of the Extracellular Matrix (ECM)

Once released from cells, exosomes can become integral components of the ECM, allowing cells to manipulate ECM composition and function. During osteogenesis, exosomes with a membrane rich in phosphatidylserine (PS) play a role in initiating the formation of hydroxyapatite crystals, which are essential for bone structure. Additionally, exosomes store calcium through annexins and generate phosphate through various enzymes and transporters, further contributing to the mineralization process during osteogenesis [26,27]. Tissue factor-containing exosomes provide another example of exosome-mediated modulation of the ECM. These exosomes serve as templates for initiating thrombus formation, playing a crucial role in both routine wound repair and abnormal clotting, such as paraneoplastic clotting [28,29,30]. Exosomes also act as templates for generating diffusible signaling molecules, including adenosine, prostaglandins, plasminogen activators (PAs), and lysophospholipids [15,16,31]. These molecules can have diverse effects on cell signaling and the ECM environment. Additionally, exosomes play a role in the development, expansion, and dissemination of amyloid-rich aggregates, plaques, and tangles seen in neurodegenerative disorders [32,33,34]. They contribute to the pathological processes seen in conditions such as Alzheimer’s disease (AD), where amyloid plaques and neurofibrillary tangles are hallmarks of the disease.

### 3.3. Intercellular Communication and Molecular Transfer

Exosomes function as signaling particles, capable of delivering multiple signals through the engagement and clustering of specific receptors on the surface of recipient cells in various pathways involved in growth factors, cytokines, Wnt, and Notch. Exosomes can also fuse with target cells, leading to the transmission of functional receptors and even activated receptors and effectors. While the physiological importance of exosome-mediated signaling and intercellular traffic is still being explored in many systems, there are instances where its significance has been established. For example, exosome-mediated PD-L1 signaling has been implicated in immunosuppression [35,36,37], exosome-mediated miRNA reprogramming during pregnancy [38], viral infections [39], and the transfer of EGFRvIII in cancer [40].

## 4. Potential Therapy Tools

Due to their ability to facilitate cell-to-cell communication, exosomes are being evaluated for repurposing as carriers of therapeutic cargo in diseases such as cardiomyopathies [41], cancer [17], and neurodegenerative disorders [42]. Exosomes can be derived from cellular sources like bone marrow and mesenchymal stem cells, or they can be re-engineered to carry selected cargo such as drugs or therapeutic proteins [43]. One of the significant advantages of using exosomes as therapy tools is their ability to transport cargo across the blood–brain barrier (BBB) and deliver it to the brain. This makes them particularly promising for addressing neuropathologies. Exosomes have been confirmed to be secreted by various cell sources in the central nervous system including neurons [44], microglia [45], astrocytes [46], oligodendrocytes [47], neural stem cells [48], and cerebrospinal fluid [49]. Exosomes, small vesicles derived from cells, carry a distinctive biological cargo reflecting the source cell’s health. This cargo offers valuable insights for studying conditions and potential therapeutic interventions. Exosomes have significant potential as therapy tools, transporting various substances across barriers. However, further research is essential to fully harness their therapeutic capabilities, particularly in treating neurological disorders. Exosomes’ ability to traverse the BBB makes them attractive for neurological disease treatment. Research indicates the importance of optimizing isolation techniques as different methods can impact exosome size, potentially influencing their therapeutic potential. The mechanism of exosome uptake into target cells is crucial for evaluating their effectiveness as nanotherapeutics, and engineering exosomes with specific “zip codes” can enhance delivery efficiency. Exosomes, unlike soluble factors, can deliver large amounts of cargo directly to recipient cells, enabling more efficient and targeted therapies. In vivo administration of exosomes has shown promising results, including reducing apoptosis [50], modulating inflammation [51], inhibiting cancer growth [52], and enhancing myocardial viability [53]. Compared to synthetic vectors, exosomes exhibit greater bioavailability and inherent biological properties, making them promising for therapy [54]. The unique properties of exosomes hold potential for overcoming challenges in drug delivery, such as crossing the BBB, making them valuable for therapeutic applications. (Figure 3). While exosome therapy holds promise, several barriers must be overcome for widespread implementation. These involve assessing loading capacity and cargo half-life, understanding pharmacological considerations for dosage and biodistribution, and studying the kinetics of exosome uptake by target cells. Comprehensive research into bioengineered ligands on exosome surfaces for improved delivery to specific targets is crucial for developing effective therapeutic strategies.

Exosomes show promise in treating various neurodegenerative conditions like stroke [55], ALS [56,57], Alzheimer’s [58], dementia [59,60], and Parkinson’s [61]. Mesenchymal stem cell (MSC)-derived exosomes exhibit tissue-protective effects in stroke models [62] and enhance neurite growth, potentially through microRNA transfer aiding neuronal recovery in spinal cord injury [63]. In neuroinflammatory diseases like ALS, exosomes influence immune responses [64]. Studies in ALS reveal exosomal mediation of TDP-43 infiltration into the central nervous system, contributing to increased neuroinflammation and neurodegeneration [65]. Exosomes are considered potential therapeutic targets for ALS. In Alzheimer’s, exosomes are explored for their therapeutic potential, with mesenchymal stem cell-derived exosomes delivering neprilysin to degrade beta-amyloid peptides, potentially alleviating Alzheimer’s symptoms [66] (Figure 4).

Additionally, in mouse models, dendritic cell-derived exosomes loaded with short interfering RNA (siRNA) have been used to target AD [67]. These exosomes, equipped with siRNA, can cross the BBB and knock down the mRNA and protein levels of BACE1, a protease involved in the production of amyloid precursor proteins [14]. This targeted exosomal delivery of siRNA has the potential to alleviate the pathogenesis of AD (Figure 5).

Early studies emphasize the emerging role of exosomes in neurological disorders, presenting them as potential therapeutic targets for ALS and AD. Frontotemporal dementia (FTLD), marked by frontal and temporal brain lobe degeneration, often occurs between ages 40 and 65. Mutations in the progranulin gene (GRN), a key cause of FTLD, lead to decreased exosome-associated progranulin proteins, impacting cell-to-cell communication [68]. Null GRN mutations reduce exosome release, lowering brain progranulin levels and contributing to FTLD neurodegeneration. Addressing this altered exosomal communication is considered a potential therapeutic target for dementia causes. In Parkinson’s disease, characterized by neuroinflammation and dopamine neuron loss, exosome-centered approaches are explored. Mutations in the leucine-rich repeat kinase 2 (LRRK2) gene are linked to inherited PD [69]. Recent research finds elevated autophosphorylated LRRK2 in urinary exosomes, correlating with cognitive impairment severity [70,71]. Exosomal delivery of LRRK2 mutations emerges as a specific therapeutic target, and the detection of LRRK2-containing exosomes in urine offers a potential biomarker for this Parkinson’s form. Misfolded proteins, implicated in ALS, Alzheimer’s, and Parkinson’s, connect various neurodegenerative diseases. Exosomes are capable of transmitting these misfolded proteins, which can further contribute to the pathology of these diseases. Moreover, exosomes can facilitate the aggregation of proteins, thereby accelerating neurodegeneration by transmitting aggregates to previously unaffected regions of the central nervous system (CNS).

### 4.1. Noninvasive Delivery of Exosomes to the Brain

The intricacy of brain diseases, coupled with the absence of proficient technologies for drug delivery across the BBB, poses a substantial hurdle to CNS drug development. The BBB presents a formidable challenge as only small molecules with lipid solubility and a molecular weight less than 400 Da can effectively traverse it, leaving the majority of macromolecules unable to penetrate the brain endothelium. This physiological barrier at the BBB restricts entry for 95% of molecules considered for drug development. Historically, this persistent challenge of drug delivery to the brain [72] has led to the abandonment of further development for specific therapeutic agents, as achieving sufficient drug levels in the brain through systemic circulation remains difficult [72]. A pioneer study used an innovative approach to address this challenge by utilizing exosomes for noninvasive intranasal drug delivery to the brain. The study demonstrated the therapeutic potential of exosome-complexed curcumin in lipopolysaccharide (LPS)-induced inflammation and experimental allergic encephalomyelitis, as well as an exosome-complexed Stat3 inhibitor in a glioblastoma tumor model [73]. Although curcumin and Stat3-targeting inhibitors have proven efficacious in treating various inflammatory neurological diseases, their clinical use has been hampered by poor bioavailability and the inability to traverse the BBB [74,75]. The findings suggest that intranasally administered exosomes may serve as effective carriers for therapeutic agents, enhancing their biological stability and facilitating their bypass of the BBB (Figure 6).

Lately, exosomes have garnered significant interest as promising drug delivery vehicles for treating CNS diseases. The initial clinical studies utilizing autologous exosome-based therapies have demonstrated favorable tolerability [76,77,78]. While these studies are promising for employing autologous exosomes as personalized medicines, certain issues must be addressed to render them applicable for therapeutic purposes in CNS diseases.

Exosomes exhibit several inherent characteristics that render them highly suitable for serving as carriers in drug delivery. Their biocompatibility, stability in circulation due to size, endogenous origin, and surface composition [79], along with the ability to evade phagocytosis and the immune system [80,81], make them advantageous. Additionally, exosomes can efficiently deliver their cargo by interacting with target cells through receptor binding or direct membrane interaction [82]. It is possible to engineer tissue-type-specific or cell-type-specific targeting ligands onto their surface [83], and exosomes can be loaded with various cargos, including nucleic acids, proteins, and low-molecular-weight therapeutic agents [12]. Furthermore, the potential for personalized medication is offered through the re-administration of isolated exosomes loaded with therapeutic agents into the patient [84,85]. However, numerous challenges must be addressed to render exosomes applicable for therapeutic purposes in the treatment of central nervous system diseases.

### 4.2. Role of Exosomes and Therapeutic Potential in Alzheimer’s Disease (AD)

AD is the most common neurodegenerative disease and cause of dementia in the elderly, and the role of EVs has gained significant attention in AD research. EVs are involved in the intercellular communication and transfer of bioactive molecules. The altered secretion and functions of EVs have been observed during the progression of AD, and blocking EV release has been shown to mitigate AD symptoms. The roles of neuron-derived EVs (NDEVs) [86], astrocyte-derived EVs (ADEVs) [87], and microglia-derived EVs (MDEVs) [88] in AD pathology have been investigated. NDEVs released by neurons have been found to carry AD-related factors such as phosphorylated Tau protein (p-Tau) and amyloid-beta (Aβ) [89]. Studies have shown that NDEVs facilitate the spread of AD-related factors among brain cells and contribute to the formation of Aβ amyloid fibrils in the CNS [90]. Additionally, NDEVs in AD demonstrate dual effects, indicating dynamic functional changes during AD progression. Astrocyte-released ADEVs, abundant in the CNS, contribute to AD pathogenesis by promoting Aβ aggregation and impeding its uptake by neuroglia, ultimately leading to neuronal loss [87]. These vesicles transfer Aβ processing enzymes, pro-inflammatory factors, and p-Tau, contributing to Aβ deposition and neurotoxicity [91]. However, it is also speculated that ADEVs may have beneficial effects on AD, although further investigation is needed. MDEVs released by microglia, the resident immune-competent cells of the brain, play a role in neuroinflammation and the spread of AD pathogenic factors [88]. MDEVs have been shown to transfer Aβ and tau among cells, leading to increased Aβ neurotoxicity and abnormal tau aggregation [92,93]. Additionally, MDEVs exhibit altered protein and lipid profiles in AD, suggesting their involvement in neuroinflammation and neurotoxicity [94,95]. However, MDEVs derived from M2-polarized microglia have shown beneficial effects by restoring neuronal viability, reducing Aβ deposition, and facilitating Aβ phagocytosis [96,97,98].

### 4.3. Role of Exosomes and Therapeutic Potential in Parkinson’s Disease (PD)

The degeneration of dopaminergic neurons in the substantia nigra of the midbrain leads to a decrease in dopamine levels in the striatum [99]. The spreading of the neuronal protein α-synuclein (α-syn) with pathological conformations through EVs has emerged as a key factor in the degeneration of dopaminergic neurons [99]. NDEVs have been found to contain α-syn in in vitro PD models and in the plasma of PD patients [100]. These NDEVs can transfer α-syn to recipient cells, including neurons and microglia, promoting α-syn aggregation and neuroinflammation [101]. Additionally, NDEVs have been shown to carry other PD-related proteins [102] and miRNAs [103] associated with autophagy regulation and inflammatory responses, further contributing to the onset and progression of PD. However, NDEVs have also demonstrated beneficial effects, such as reducing neuroinflammation in animal models of PD [104]. ADEVs are involved in the clearance of extracellular α-syn and can induce an inflammatory response in astrocytes [105,106]. Although the presence of α-syn in ADEVs and their direct involvement in α-syn spreading remain unclear, alterations in ADEV cargo enrichment, such as specific astrocyte-derived exosomal microRNA-200a-3p miRNAs, can contribute to the progression of PD [107].

MDEVs have been shown to contain α-syn oligomers and can transfer them to neurons, leading to dopaminergic neuron degeneration [108]. α-syn can induce an increase in exosomal secretion by microglia, creating a cycle that exacerbates MDEV-mediated spreading of α-syn [109]. Inflammatory conditions, such as α-syn/interferon-γ (IFN-γ)/LPS stimulation, can further enhance MDEV-mediated dopaminergic neurodegeneration [109].

Oligodendrocyte-derived EVs (ODEVs) have recently been investigated in the context of PD. Studies have demonstrated higher levels of ODEVs, containing α-syn, in the plasma of PD patients compared to healthy controls [110]. However, our understanding of the pathological contributions of ODEVs in PD is limited, and further research is needed.

### 4.4. Role of Exosomes and Therapeutic Potential in Amyotrophic Lateral Sclerosis (ALS)

Different types of EVs, including neuronal-derived EVs (NDEVs), ADEVs, and MDEVs, have been found to play pathological roles in ALS. NDEVs have been shown to deliver pathogenic factors to neuroglial cells. Studies have reported that NDEVs from ALS mouse models contain misfolded neurotoxic proteins, such as mutant SOD1, dipeptide repeat proteins (DPRs), and TAR DNA-binding protein-43 (TDP-43) [111,112]. These pathogenic factors can be internalized by microglia and astrocytes, inducing inflammatory responses and neurodegeneration [113]. Altered miRNA profiles have also been observed in NDEVs isolated from the plasma of ALS patients [114], suggesting a potential role of NDEV-mediated miRNA dysregulation in ALS pathogenesis. ADEVs from ALS models have shown altered content profiles, including the packaging of mutant SOD1 [115]. The delivery of mutant SOD1 from astrocytes to neurons via ADEVs can induce selective motor neuron death [115]. ADEVs derived from ALS patients with C9orf72 mutations have been found to lack miR-494-3p, a negative regulator of axonal maintenance-related gene SEMA3A, potentially contributing to motor neuron degeneration [116]. Additionally, ADEVs isolated from the plasma of sporadic ALS patients have shown increased levels of pro-inflammatory factors like IL-6, suggesting a role in the pathological spread of neuroinflammation in ALS [117]. MDEVs, derived from activated microglia in ALS, have been implicated in the release of mutant SOD1 and pro-inflammatory molecules [118]. Mutant SOD1 is released via MDEVs [119] and taken up by motor neurons, leading to neurotoxicity [119]. Increased levels of HMGB1 [120], miR-155 [121], and miR-146a [122] have been observed in MDEVs, contributing to neuroinflammation and exacerbating ALS phenotypes.

### 4.5. Role of Exosomes and Therapeutic Potential in Huntington’s Disease (HD)

EVs play a role in HD progression, as evidenced by elevated levels of total huntingtin (HTT) in plasma-derived EVs from HD patients [123]. Neuronal EVs (NDEVs) carrying mRNA with an expanded CAG-repeat element suggest a potential involvement in spreading pathogenic HTT within the brain [124]. While total HTT and mutant HTT fragments are not detected in NDEVs, in vitro studies suggest a role in suppressing HTT levels through the transfer of HTT-targeting microRNAs [125]. Further research is needed to fully comprehend NDEVs’ pathological or beneficial roles in HD. Astrocytic EVs (ADEVs) play a less-explored role in HD pathogenesis. The sequencing of highly expressed genes in ADEVs indicates their involvement in promoting HD [126]. In HD mouse models, mutant HTT hampers ADEV release, disrupting αB-Crystallin sorting into ADEVs. This disturbance triggers neuroglial activation, neuroinflammation, and subsequent neurodegeneration in HD [127]. In summary, both NDEVs and ADEVs are implicated in the pathological processes of HD. NDEVs may participate in the spreading of pathogenic HTT within the brain and have the potential to suppress HTT levels through the transfer of HTT-targeting microRNAs. ADEVs, on the other hand, may promote HD by causing neuroinflammation and neurodegeneration through disrupted EV release and impaired sorting of CRYAB (HspB5 or αB-Crystallin).

### 4.6. Role of Exosomes and Therapeutic Potential in Stroke

Studies using MSC-derived exosomes show promise in treating stroke [128,129], traumatic brain injury (TBI) [130], and intracerebral hemorrhage (ICH) [131] in animal models. The intravenous administration of exosomes improves neurovascular remodeling and enhances neurological recovery [129,132] (Figure 7).

In fact, the effects of MSC-derived exosomes were comparable to those observed with MSC therapy itself. A head-to-head comparison between MSCs and MSC-derived exosomes in a stroke model showed equal improvements in motor function and coordination [133]. Additionally, MSC-derived exosomes have been shown to improve sensorimotor function in a rat model of subcortical stroke and preserve pattern separation and spatial learning ability in a mouse model of TBI [134].

These findings suggest that MSC-derived exosomes have the potential to mimic the therapeutic effects of MSC therapy in neurological disorders. The use of exosomes as a therapy holds promise due to their ability to transfer bioactive molecules and mediate intercellular communication. More research is essential to grasp exosomes’ mechanisms and optimize their clinical application. Few studies have shown the positive effects of MSC-derived exosomes in improving sensorimotor and cognitive function in various animal models of neurological disorders. In a rat model of intracerebral hemorrhage (ICH), treatment with MSC-derived exosomes led to improvements in sensorimotor and cognitive function [131]. Similar benefits have been observed in large animal models, such as sheep and pigs, where the administration of exosomes derived from MSCs resulted in reduced seizure frequency, preserved reflex sensitivity, and promoted neurological recovery after brain injury [135].

Furthermore, studies in adult rhesus monkeys with cortical lesions demonstrated that treatment with MSC-derived exosomes led to a faster and more complete recovery of fine motor function, comparable to that of uninjured animals [136]. These findings suggest that MSC-derived exosomes play a role in mediating the therapeutic effects of MSC therapy in neurological disorders. Preclinical and human safety studies affirm that MSC-derived exosomes are well-tolerated without causing adverse immune reactions [137,138]. Efficacy is demonstrated in rat traumatic brain injury models and a human graft versus host disease case [139]. Exosomes from various cell types show potential in stroke-induced brain remodeling [140], yet their effects on immune responses and long-term neuroprotection need further exploration. Research indicates promising therapeutic potential for neurological disorders, promoting functional recovery and neuroprotection [141]. In acute ischemic stroke, exosomes, particularly from human cardiosphere-derived cells, enhance outcomes when administered with tissue plasminogen activator [142]. The most effective cell-derived exosomes for stroke and traumatic brain injury treatment remain unknown [143], with transcriptomic and proteomic analyses suggesting functional equivalence between embryonic and bone marrow-derived MSCs [144]. Exosome content is determined by the parent cell, warranting investigation into the comparability of cargo from induced pluripotent stem cell-derived MSCs and adult MSCs [145].

MSC-derived exosomes show promise for stroke and TBI therapy, offering benefits over MSCs by avoiding safety concerns [146,147]. Naive exosomes from healthy MSCs provide therapeutic effects without the adverse outcomes observed with MSCs. In neurological disorders, endogenous exosomes exacerbate inflammation, impacting stroke prognosis. However, MSC-derived exosomes reduce inflammation in preclinical studies, mitigating brain injury and immunosuppression. The therapy’s impact extends beyond the brain, affecting heart function and aging speed, highlighting the systemic role of exosomes. The inhibition of choroid plexus epithelium exosome secretion attenuates peripheral-induced brain inflammation. Overall, exosomes demonstrate a complex yet promising role to modulate inflammation locally and systemically, making them potential therapeutic targets for neurological disorders.

### 4.7. Cerebral Aneurysm

A cerebral aneurysm is a bulge in a brain artery that [148] when ruptured, poses severe health risks like hemorrhagic stroke, brain damage, coma, or death [149]. Symptoms may include pain, numbness, weakness, facial paralysis, dilated pupils, or vision changes. A ruptured aneurysm manifests as a sudden, severe headache with symptoms like double vision, nausea, vomiting, a stiff neck, sensitivity to light, seizures, and loss of consciousness. Immediate medical attention is crucial if individuals experience a sudden, severe headache with other symptoms. Understanding the interplay between exosomes and cerebral aneurysms may offer new insights into diagnostic and therapeutic strategies for this condition.

### 4.8. Formation, Risk Factors, and Rupture Risk

Cerebral aneurysms form when the walls of brain arteries weaken, often occurring at branch points due to vulnerability. While they may be congenital [150], risk factors include genetic connective tissue disorders [151], polycystic kidney disease [152], arteriovenous malformations [153], and a family history of aneurysms [154]. Other factors like high blood pressure, smoking, drug abuse, age over 40, head trauma, brain tumors, and infections in arterial walls contribute [155]. Hypertension, smoking, diabetes, and high cholesterol increase the risk of atherosclerosis, leading to fusiform aneurysms. Rupture risk factors involve smoking, high blood pressure, aneurysm size, location (posterior communicating and anterior communicating arteries), growth, and family history. Individuals with multiple aneurysms, previous rupture, or sentinel bleed face the highest risk [156].

Cerebral aneurysms vary in their need for treatment. Small, unruptured aneurysms, especially those without associated risk factors, may be safely monitored without immediate intervention. The aggressive treatment of coexisting medical problems is essential [157]. Treatment decisions for unruptured aneurysms depend on factors like type, size, location, rupture risk, age, health, and medical history [158]. Treatments must be considered carefully due to potential serious complications. Rupture risk may be reduced through blood pressure control, smoking cessation, and avoiding stimulant drugs. Surgical or endovascular options may be recommended to manage symptoms and prevent damage. However, these options carry risks like blood vessel damage, aneurysm recurrence, rebleeding, and the possibility of stroke.

Many aneurysmal subarachnoid hemorrhage (SAH) patients who undergo treatments like clipping or coiling may still face cognitive function impairments, impacting their ability to return to work [159]. The severity of bleeding is linked to neuropsychological outcomes [160]. Despite efforts with anti-vasospasm therapy, outcomes in SAH patients have not significantly improved, leading to an increased focus on early brain injury (EBI) within the first 72 hours post-SAH as a critical risk factor for poor outcomes [161,162]. EBI plays a role in diverse pathophysiological changes post-SAH, encompassing global cerebral edema, ultra-early vasospasm, and reactive neuroinflammation [163]. Consequently, EBI is considered a fundamental contributor to cognitive dysfunction post-SAH, emphasizing the need for targeted interventions to enhance the overall outcomes of SAH patients [162,164]. Recent research has highlighted the potential role of exosomes in mediating neuroinflammation and contributing to the pathophysiological changes post-SAH. Accumulating evidence suggests that exosomes released from damaged brain cells could propagate neuroinflammatory responses and exacerbate early brain injury.

### 4.9. Exosomal Non-Coding RNAs: Diagnostic and Therapeutic Potential in Intracranial Aneurysms and Aneurysmal Subarachnoid Hemorrhage (SAH)

The exploration of exosomal nucleic acids has evolved significantly since the initial report [13]. Exosomes play a crucial role in cell interactions, involving the translation of exosomal mRNA in recipient cells and the regulation of target genes through various exosomal nucleic acids, such as mRNAs, non-coding RNAs, and DNAs [165,166,167]. Within this spectrum, non-coding RNAs derived from exosomes, such as miRNAs, lncRNAs, and circRNAs, have emerged as particularly prominent subjects of study, especially in cerebrovascular diseases (CVDs) [18,168,169,170,171].

MiRNAs, measuring around 17–24 nucleotides, constitute a crucial component of the intricate gene regulatory network governing cell processes like proliferation, differentiation, and apoptosis [172]. These small RNAs, actively released by cells, exhibit remarkable stability during circulation in various forms [173,174,175]. Exosomal miRNAs, easily detectable and extractable, are gaining prominence as diagnostic indicators for CVDs [176,177]. Beyond their role as biomarkers, miRNAs have demonstrated involvement in the early protection and recovery of the nervous system after stroke [168].

Mesenchymal stem cell (MSC) transplantation has emerged as a promising therapeutic approach for subarachnoid hemorrhage (SAH) in rodent models [178]. The therapeutic potential of MSCs in SAH was demonstrated by the intravenous administration of MSCs improving structural integrity and promoting functional recovery in rats subjected to SAH [179,180,181]. A study demonstrated that administering bone marrow MSC (BMSC) alleviated early brain injury (EBI) induced by SAH and reduced neurobehavioral impairments, partly through inhibiting microglia activation [178].

Despite the potential benefits, the use of cell-based therapy, including MSC transplantation, raises concerns about potential risks. These risks encompass the development of tumors or immune rejection [182,183]. Therefore, while MSC transplantation shows therapeutic promise for aneurysmal SAH in rodent models, careful consideration of potential risks is essential for translating these findings into clinical applications [184].

Several studies have indicated that the therapeutic benefits of mesenchymal stem cell (MSC) therapy may be attributed to EVs released by MSCs, as opposed to the transdifferentiation of MSCs themselves [185,186]. Notably, the therapeutic impact of MSC-derived EVs has been demonstrated to be comparable to that of MSCs in animal models of ischemic stroke and traumatic brain injury (TBI) [187]. In a study investigating the protective role of mesenchymal stem cell-derived extracellular vesicles (MSC-EV) in a rat model of subarachnoid hemorrhage (SAH), significant improvements were observed in the neurological scores and mitigation of brain edema. MSC-EV administration resulted in the uptake of EVs by damaged neurons, confirming their role in exerting therapeutic effects. Assessment through a TUNEL assay and cleaved-caspase-3/NeuN double staining indicated a reduction in apoptosis in the SAH + EV group compared to the SAH + PBS group [188].

At the molecular level, MSC-EV increased Bcl-2 expression and inhibited Bax and Caspase-3 expressions in both the prefrontal lobe and hippocampus. This suggests that MSC-EV exerts its neuroprotective effects by suppressing cellular apoptosis through the inhibition of the mitochondrial apoptosis pathway. To delve into the underlying mechanisms, miRNA EV sequencing was employed to identify specific molecular cargo. Based on data from exRNA Atlas, it was hypothesized that miR-21-5p plays a pivotal role as the key mediator of the therapeutic effects of MSC-EV in SAH. miR-21-5p inhibitors and MK2206 were employed to investigate the underlying mechanism, with both demonstrating the capability to diminish the effects of MSC-EV. The research revealed that SAH rats receiving MSC-EV treatment displayed enhanced cognitive function in the Morris water maze test compared to those treated with PBS [188]. This comprehensive analysis provides insights into the multifaceted neuroprotective mechanisms of MSC-EV in SAH, linking improvements in neurological outcomes to the modulation of apoptotic pathways and highlighting the potential involvement of specific miRNAs.

### 4.10. Harnessing Exosomal Biomarkers for the Early Detection and Monitoring of Cardiovascular Diseases and Intracranial Aneurysms

Angiographic monitoring is vital for early CVD diagnosis and treatment. However, it is unsuitable for routine screening, and current assessment relies heavily on symptoms and imaging. Exosomes, stable in body fluids and easily extractable, offer convenient and sensitive indicators, reflecting cell pathology and providing early insights into abnormal cargo before observable organ changes. These characteristics position exosomes as promising biomarkers for identifying and monitoring CVDs, offering a potential breakthrough in diagnostic approaches [165,189].

Intracranial aneurysms (IAs), characterized by abnormal arterial wall dilation, pose a risk of aneurysmal SAH. Monitoring IA formation and growth through imaging is challenging, but changes in gene expression and exosomal non-coding RNAs, particularly elevated miR-29a-3p and miR-145-5p [190] in the plasma exosomes of IA patients, may serve as early warning signs. Notably, higher miR-145-5p levels in ruptured aneurysms than in unruptured ones suggest distinct molecular mechanisms between IA growth and rupture. Despite clear aSAH evidence via computed tomography and lumbar puncture, false negatives with atypical features may risk rebleeding. A recent study employed least absolute contraction and selection operator (LASSO) analysis to develop a 24-miRNA classifier, demonstrating high accuracy and specificity in discriminating SAH from parenchymal hemorrhage [191]. Differentially expressed plasma exosomes may provide insights into distinct etiologies and contribute to the diagnosis of aSAH [18].

A recent study investigated the in vitro effects of blood-cerebrospinal fluid (BCSF) on brain microvascular endothelial cells (BMECs) and explored the expression of exosomal miR-630 in the cerebrospinal fluid (CSF) obtained from patients three days after experiencing aneurysmal SAH [192]. The findings indicate that BCSF treatment resulted in substantial reductions in the mRNA expressions of key molecules, including ICAM-1, VCAM-1, and ZO-1, in BMECs. Additionally, the production of nitric oxide by BMECs was diminished following exposure to BCSF. The in vitro BCSF-treated group also exhibited a significantly lower expression of exosomal miR-630. Furthermore, in CSF collected from SAH patients at the three-day mark, miR-630 expression was found to be markedly reduced by 3.77-fold compared to control patients. The study further investigated the impact of miR-630 on BMECs by co-culturing with exosomes containing miR-630 demonstrating a substantial increase in the expressions of ICAM-1, VCAM-1, and ZO-1.

The findings indicate the regulatory influence of BCSF and alterations in exosomal miR-630 expression on crucial molecules involved in endothelial cell adhesion and barrier function. The study suggests a potential role for miR-630 in modulating the response of BMECs, offering valuable insights into the pathophysiological mechanisms underlying aneurysmal SAH. As an early study has shown the role of brain microcirculation in early brain injury (EBI) following subarachnoid hemorrhage (SAH) [193], the potential role of miR-630 in protecting brain microvascular endothelial cells (BMECs) opens avenues for therapeutic and prophylactic interventions targeting post-SAH vasospasm and cerebral ischemia.

### 4.11. Macrophage-Derived Exosomes and Their Implications in Aneurysmal Pathology

Intracranial aneurysm (IA) pathogenesis involves various changes, such as endothelial dysfunction, internal elastic lamina disruption, VSMC apoptosis, phenotypic transformation, extracellular matrix degradation, and chronic inflammation. While macrophage infiltration was conventionally considered the primary driver of aneurysmal wall inflammation, recent insights, including the discovery of macrophage-derived exosomes in abdominal aortic aneurysm (AAA) tissue, challenge this perspective. Both human and murine AAA models show the presence of these exosomes. Moreover, macrophage-derived exosomes correlate with an upregulation of matrix metalloproteinases 2 (MMP2) expression in cultured VSMCs. These combined in vivo and in vitro findings indicate the significant role of macrophage-derived exosomes in the pathological processes of aneurysms [194].

### 4.12. Exosome-Mediated Regulation of the Vascular Smooth Muscle Cell Phenotype in Intracranial Aneurysms

The intricate interplay between exosomes and vascular smooth muscle cells (VSMCs) has emerged as a crucial factor in the progression of intracranial aneurysms (IAs). Recent research indicates that tumor-associated macrophages (TAMs) carrying miR-155-5p play a pivotal role in stimulating IA formation [195]. This occurs through the promotion of VSMC proliferation and migration, along with the activation and infiltration of TAMs. This study sheds light on the significant impact of exosome-mediated phenotypic transformation in driving IA progression.

Moreover, evidence from studies on other vascular diseases suggests that exosomes derived from endothelial cells (EC-Exos) influence the VSMC phenotype. EC-Exos have been shown to stimulate the expression of vascular cell adhesion molecule-1 (VCAM-1) and upregulate pro-inflammatory molecules in VSMCs [196]. A recent study directly showed that decreased TET2 in EC-Exos leads to VSMC phenotypic switching and contributes to intimal hyperplasia post-arterial injury [197]. Conversely, exosomes derived from endothelial progenitor cells (EPC-Exos) have been found to inhibit the synthesis phenotype of VSMCs, indicating a complex regulatory network mediated by exosomes [198]. These findings underscore the dynamic role of exosomes in modulating the VSMC phenotype, providing valuable insights into the underlying mechanisms of IA progression and potential therapeutic targets for intervention.

### 4.13. MSC-Exos in Intracranial Aneurysm Therapy

Mesenchymal stem cells (MSCs) have emerged as promising players in the inhibition of pathological processes associated with intracranial aneurysms (IAs) and have been actively explored in cell therapy for IAs [199]. Notably, the therapeutic potential of MSCs extends to their exosomes, known as MSC-derived exosomes (MSC-Exos), which inherit the therapeutic capabilities of their parent cells. In the context of IA treatment, MSC-Exo miR-144-5p demonstrates a compelling impact by promoting the proliferation of endothelial cells (ECs) and fibrocytes, ultimately enhancing EC viability in vitro. This anti-IA effect is believed to be mediated through the targeting of phosphatase and tensin homolog (PTEN) proteins [200]. The anti-inflammatory properties of MSC-Exos represent another crucial mechanism in countering IA development. Exosomal miR-147, derived from MSCs, exhibits a capacity to attenuate abdominal aortic aneurysm (AAA) formation by inhibiting macrophage activation [201]. Similarly, MSC-Exos contribute to preventing IA rupture, partially attributed to a reduction in cytokine release, as well as the modulation of tryptase and chymase activities in mast cells [202]. Furthermore, the inhibition of IA formation has been associated with the action of bone marrow mesenchymal stem cell (BMSC)-derived miR-23b-3p, which targets Kruppel-like factor 5 (KLF5) and suppresses nuclear factor κB (NF-κB). This process is accompanied by an upregulation of the contractile phenotype in vascular smooth muscle cells (VSMCs), highlighting the intricate relationship between inflammation and the phenotypic transformation of VSMCs [203]. These findings underscore the potential therapeutic implications of MSC-Exos in managing IAs and offer insights into their multifaceted mechanisms of action.

## 5. Future Directions

The future directions in exosome research are focused on addressing several key questions and advancing our understanding of exosome biogenesis, composition, trafficking, and functional delivery. Some of the pressing issues and areas of investigation include the following:

Assigning exosome biogenesis: Understanding how different cells distribute exosome biogenesis between their endosome and plasma membranes, and how this process influences exosome composition and heterogeneity;

Regulation of IPMCs: Investigating the mechanisms by which cells regulate the biogenesis and gating of intraluminal vesicles (ILVs) within multivesicular bodies (MVBs) in different cell types, as it impacts exosome formation;

ESCRT proteins and function: Resolving the precise contributions of ESCRT proteins (endosomal sorting complex required for transport) in exosome biogenesis, as they are involved in ILV formation within MVBs;

Physicochemical aspects of biogenesis: Exploring the positive and negative regulators that influence exosome biogenesis at the physicochemical level, including factors that promote or inhibit ILV formation;

Protein and RNA packaging: Discovering the molecular mechanisms underlying the selective packaging of proteins and RNA into exosomes, including the identification of specific proteins and protein networks involved in this process;

Exosome trafficking: Mapping the flow of exosomes from different tissues into various biofluids and target tissues, understanding the routes and mechanisms by which exosomes are released and reach their intended destinations;

Functional delivery of cargo: Investigating the mechanisms by which bioactive proteins, lipids, and nucleic acids within exosomes are delivered and exert their effects on target cells, elucidating the molecular machinery involved in this process;

In-depth studies on the dynamic role of exosomes, especially macrophage-derived and MSC-derived exosomes, in modulating the vascular smooth muscle cell phenotype, providing insights into IA progression and potential therapeutic targets.

Advancements in these areas will require the development of new experimental approaches and technologies. The ability to interrogate exosome composition at the single-vesicle level and analyze exosome biogenesis in single cells will be crucial for gaining deeper insights into the role of exosomes in health and disease. Ultimately, this knowledge can be translated into the development of exosome-based therapies and diagnostics.

## 6. Conclusions

Exosomes are now recognized to have a crucial role in both maintaining brain health and contributing to neurological diseases. In recent years, researchers have gained a deeper understanding of exosome biogenesis, their function as carriers and communicators between cells, and their role as part of the cellular secretome. These activities of exosomes allow cells to transport both beneficial and harmful molecules throughout the body. Due to their ability to carry a diverse range of cargo, exosomes are now considered excellent vehicles for delivering therapeutic molecules to previously inaccessible regions of the brain. The field of exosomes as a potential therapy for various neurological disorders is rapidly advancing. Numerous preclinical studies have demonstrated that both natural and engineered exosomes possess powerful therapeutic effects. However, the precise cellular and molecular mechanisms underlying these therapeutic benefits are still not fully understood. The field of exosomes as a therapy for neurological disorders shows great promise; further research is needed to unravel the underlying mechanisms and overcome the challenges associated with safety, manufacturing, and quality control.

## Figures and Tables

**Figure 1 cells-13-00670-f001:**
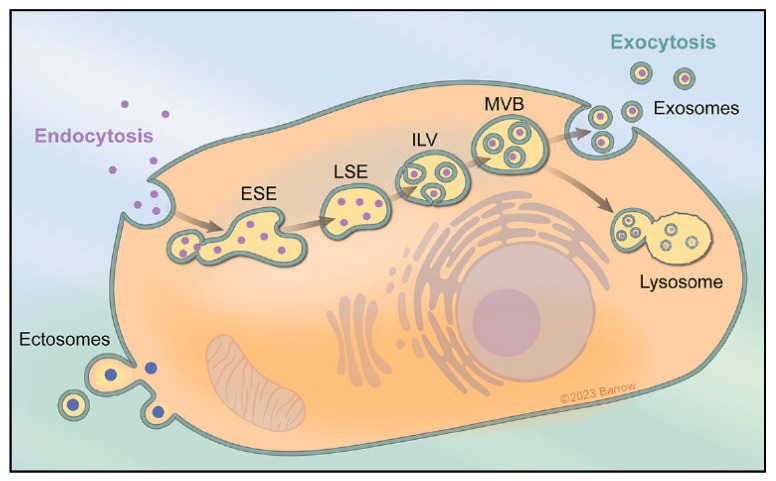
Exosomes form through the cellular uptake of extracellular components via endocytosis and plasma membrane invagination. This process initiates the creation of early-sorting endosomes (ESEs) that may bud independently or fuse with preformed ESEs from the ER, TGN, or mitochondria. Fusion with the ER and TGN allows endocytic cargo access to ESEs, progressing to the formation of late-sorting endosomes (LSEs). LSEs undergo invagination, generating intraluminal vesicles (ILVs) that include diverse constituents from various sources. ILVs of different sizes and content result based on invagination volume. LSEs evolve into multivesicular bodies (MVBs), representing future exosomes. MVBs can fuse with autophagosomes for lysosomal degradation or directly fuse with lysosomes. Alternatively, MVBs can travel to the plasma membrane via the cell’s cytoskeleton, docking with MVB-docking proteins on the luminal side. This leads to exocytosis, releasing exosomes with a lipid bilayer orientation akin to the plasma membrane, contributing to the extracellular vesicle pool. Figure is used with permission from the Barrow Neurological Institute, Phoenix, Arizona, USA.

**Figure 2 cells-13-00670-f002:**
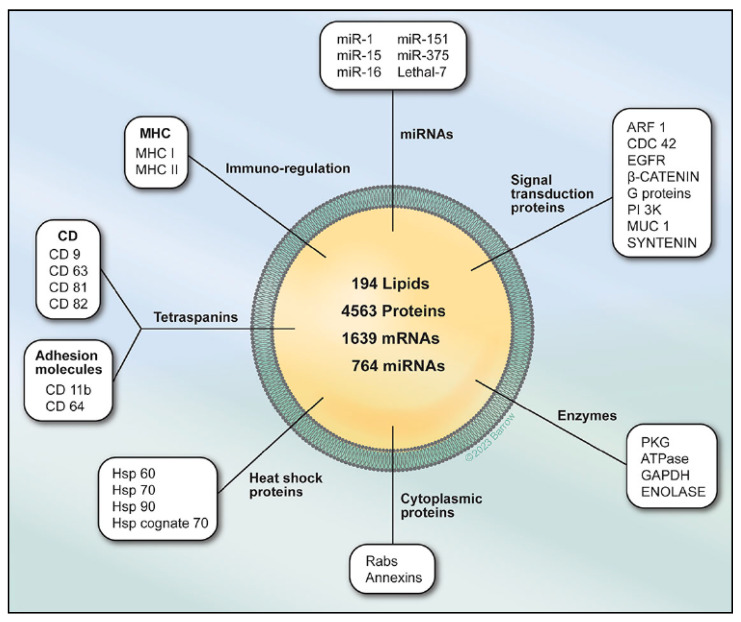
Exosomes serve as a cell-to-cell transit system within the human body, exhibiting diverse and multifunctional roles. These extracellular vesicles, produced by all cells, encapsulate nucleic acids [12,13], proteins [11,14], lipids [15,16], and metabolites [17,18]. Functioning as crucial mediators of intercellular communication, both in close proximity and over long distances, exosomes play pivotal roles in influencing various facets of cell biology. Their impact extends across health and disease, highlighting their significance in orchestrating cellular interactions and contributing to the regulation of cellular processes. Figure is used with permission from the Barrow Neurological Institute, Phoenix, AZ, USA.

**Figure 3 cells-13-00670-f003:**
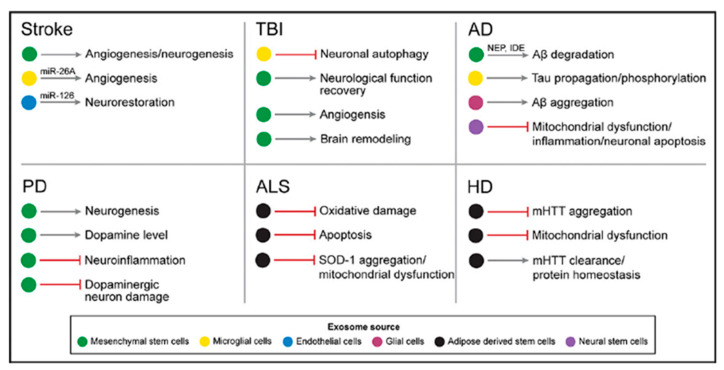
Role of exosomes in various neurodegenerative diseases. Figure is used with permission from the Barrow Neurological Institute, Phoenix, AZ, USA.

**Figure 4 cells-13-00670-f004:**
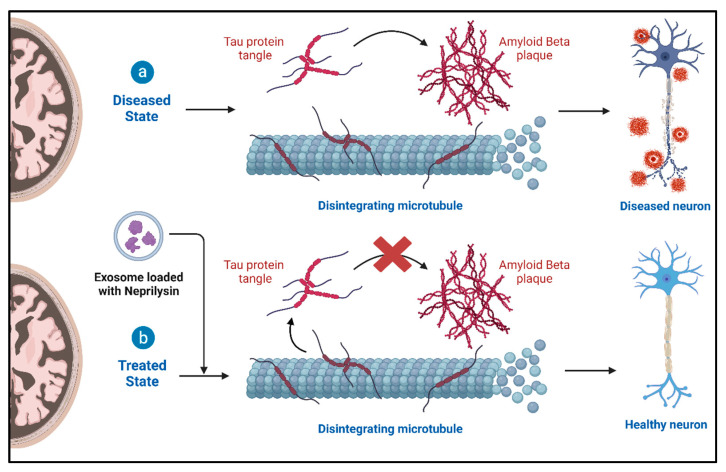
Treatment of AD with neprilysin-loaded exosomes.

**Figure 5 cells-13-00670-f005:**
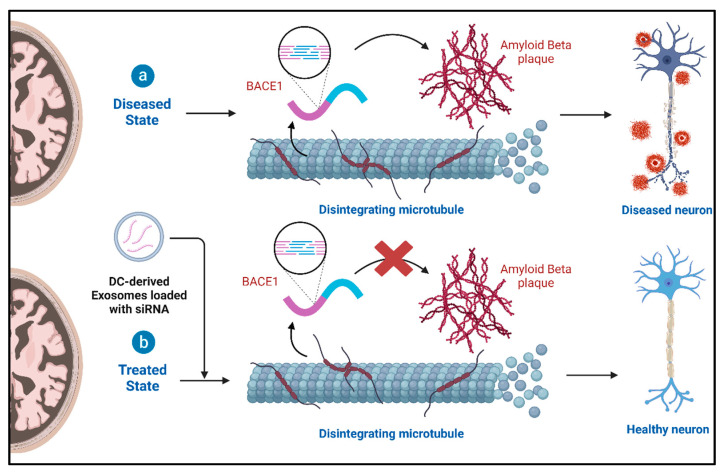
Treatment of AD with siRNA-loaded exosomes.

**Figure 6 cells-13-00670-f006:**
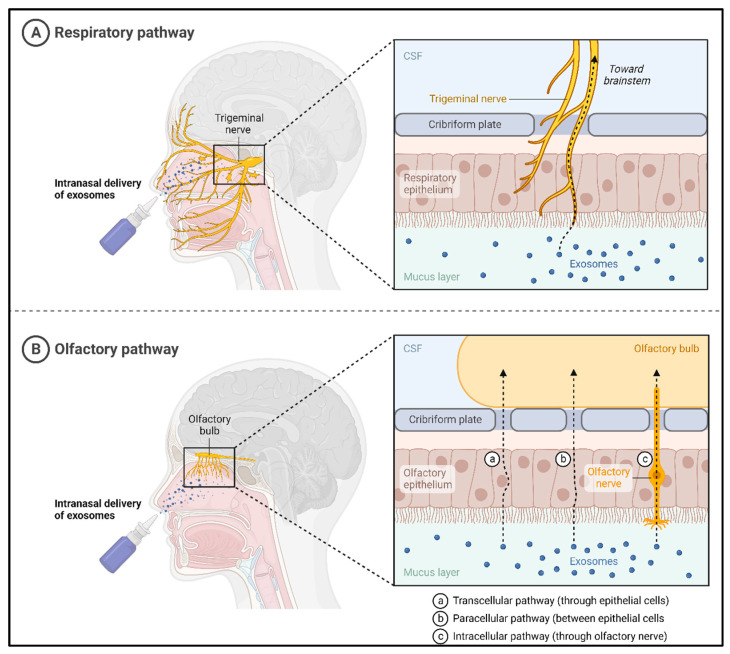
Direct pathways of nose-to-brain exosome delivery.

**Figure 7 cells-13-00670-f007:**
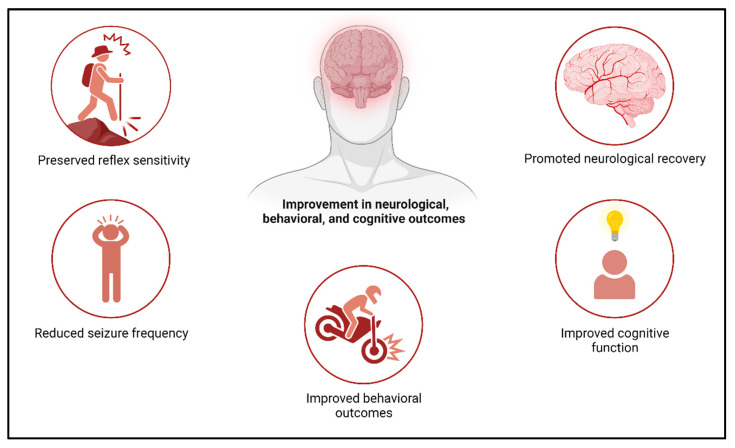
Enhancement of neurovascular remodeling after stroke and TBI by MSC-derived exosomes.

## Data Availability

Not applicable.

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
