# Peer review of "Exosomes in Vascular/Neurological Disorders and the Road Ahead"

_cells, 2024, doi:10.3390/cells13080670_

Round 1

Reviewer 1 Report

Comments and Suggestions for Authors

This is a well-written, comprehensive review of exosomes, their roles in neurological diseases, and their potential use as therapeutic agents. My comments are minor in nature and can be found below.

The manuscript should be carefully proofread to ensure it is free of grammatical and text errors. For example, in the following sentence, “Present review summarizes the roles of exosomes in neurodegenerative diseases as well as elucidating their therapeutic potential in AD, PD, ALS, HD, Stroke and aneurysm”, “The” should be placed before “present” and the first letter in “stroke” should be lowercase. Please note that these are just two examples of errors in the text and the authors are urged to carefully proofread their manuscript to ensure similar errors are corrected.

In Section 1, the authors make the distinction between exosomes and ectosomes. After describing the two vesicles, I think it would help if the authors stated that the focus of their review is on exosomes and their roles in neurological disease.

Fig. 1 requires a citation if it has been previously published. I can see a faint label inside the image “Copyright 2023 Barrow”, but a proper citation is required in the caption. This comment also applies to Fig. 2. If these images are protected by copyrights, then the authors will need to generate their own figures.

Figs. 4 and 5 should be condensed into one figure.

Sections 4.7 and 4.8 seem out of place since they take the discussion away from exosomes. I realize these two sections set the stage for Section 4.9. However, I recommend that the authors somehow find a way to weave some exosome content into Sections 4.7 and 4.8.

Section 4.9 is fairly long and should be split into at least two distinct sections.

The header in Table 1 reads “Table 11”. Please correct.

I don’t think it’s necessary to include Table 2. It could probably be removed without negatively affecting the article.

The author affiliations are difficult to interpret (e.g., confusing superscript numbers and locations). 

Comments on the Quality of English Language

None. Minor grammar check required.

Author Response

Thank you for the comments and we have modified the manuscript according to the suggestions.

Author's Reply to the Review Report (Reviewer 1)

This is a well-written, comprehensive review of exosomes, their roles in neurological diseases, and their potential use as therapeutic agents. My comments are minor in nature and can be found below.

Comment: The manuscript should be carefully proofread to ensure it is free of grammatical and text errors. For example, in the following sentence, “Present review summarizes the roles of exosomes in neurodegenerative diseases as well as elucidating their therapeutic potential in AD, PD, ALS, HD, Stroke and aneurysm”, “The” should be placed before “present” and the first letter in “stroke” should be lowercase. Please note that these are just two examples of errors in the text and the authors are urged to carefully proofread their manuscript to ensure similar errors are corrected.

Response: Thank you for bringing these errors to our attention, and we have taken care to ensure that the manuscript is free of such grammatical and text errors, as mentioned below.

  1. We believe "Stroke" should be capitalized as it is a proper noun.
  1. These ILVs" - The period after "ILVs" has been removed, as it is part of the abbreviation.
  2. "Mechnisms" has been corrected to "Mechanisms."
  3. In the phrase "dendritic cell-derived exosomes loaded with short in-terfering RNA (siRNA) have been used to target AD [63]," missing space between "in-" and "terfering." has been corrected.
  4. In the phrase "Exosomes exhibit several inherent characteristics that render them highly suitable for serving as carriers in drug delivery. Their biocompatibility, stability in circulation due to size, endogenous origin, and surface composition [76], along with the ability to evade phagocytosis and the immune system [77,78], make them advantageous," "evade" has been corrected to "evading."
  5. In the phrase "The spreading of the neuronal protein α-synuclein (α-syn) with pathological conformations through EVs has emerged as a key factor in the degeneration of dopaminergic neurons [97]," a missing space between "pathological" and "conformations." has been corrected.
  6. In the phrase "Naïve exosomes from healthy MSCs provide therapeutic effects without adverse outcomes observed with MSCs," "Naïve" has been corrected to "Naive."
  7. In the phrase "Overall, exosomes demonstrate a complex yet promising role in modulating inflammation locally and systemically, making them potential therapeutic targets for neurological disorders," "modulating" has been corrected to "to modulate."
  8. In the phrase "Overall, exosomes demonstrate a complex yet promising role in modulating inflammation locally and systemically, making them potential therapeutic targets for neurological disorders," "therapeutic" has been corrected to "therapeutics."
  9. In the phrase “The study suggest a potential role for miR-630 in modulating the response” has been corrected to “The study suggests a potential role for miR-630 in modulating the response”

Comment: In Section 1, the authors make the distinction between exosomes and ectosomes. After describing the two vesicles, I think it would help if the authors stated that the focus of their review is on exosomes and their roles in neurological disease.

Response: Thank you for the suggestion. The necessary correction has been made in Section 1 to clarify that the focus of the review is on exosomes and their roles in neurological disease.

Comment: Fig. 1 requires a citation if it has been previously published. I can see a faint label inside the image “Copyright 2023 Barrow”, but a proper citation is required in the caption. This comment also applies to Fig. 2. If these images are protected by copyrights, then the authors will need to generate their own figures.

Response: The images were produced by Barrow Neuroscience Publications (Neuropub) specifically for this review paper, and have been gratefully acknowledged for their assistance.

Comment: Figs. 4 and 5 should be condensed into one figure.

Response: We appreciate the learned suggestion to condense Figures 4 and 5 into one figure. However, each figure serves a specific purpose in illustrating different aspects of the content. Combining them may result in overcrowding and/or loss of clarity. Therefore, it would be best to keep them separate to maintain the effectiveness of each illustration.

Comment: Sections 4.7 and 4.8 seem out of place since they take the discussion away from exosomes. I realize these two sections set the stage for Section 4.9. However, I recommend that the authors somehow find a way to weave some exosome content into Sections 4.7 and 4.8.

Response: Thank you for your insightful comment. Certainly, it's important to maintain coherence throughout the document and ensure that all sections contribute to the central theme and thus have incorporated sentences as below. Furthermore, we also decide to remove table 1 and 2 to maintain the flow of the document and provide a more cohesive narrative.

“Understanding the interplay between exosomes and cerebral aneurysms may offer new insights into diagnostic and therapeutic strategies for this condition.”

“Recent research has highlighted the potential role of exosomes in mediating neuroinflammation and contributing to the pathophysiological changes post-SAH. Accumulating evidence suggest that exosomes released from damaged brain cells could propagate neuroinflammatory responses and exacerbate early brain injury.”

Comment: Section 4.9 is fairly long and should be split into at least two distinct sections.

Response: Comment is duly acknowledged and the necessary adjustments by splitting Section 4.9 into two distinct sections as below.

4.9. Exosomal Non-Coding RNAs: Diagnostic and Therapeutic Potential in Intracranial Aneurysms and Aneurysmal Subarachnoid Hemorrhage

4.10. Harnessing Exosomal Biomarkers for Early Detection and Monitoring of Cardiovascular Diseases and Intracranial Aneurysms

Comment: The header in Table 1 reads “Table 11”. Please correct.

Response: During the review process, the table was found to be unnecessary and thus the whole table has been removed from the edited manuscript.

Comment: I don’t think it’s necessary to include Table 2. It could probably be removed without negatively affecting the article.

Response: We agree and thus the table has been removed from the edited manuscript.

Comment: The author affiliations are difficult to interpret (e.g., confusing superscript numbers and locations). 

Response: Thank you so much for the comment. The necessary changes have been made to author affiliations.

Reviewer 2 Report

Comments and Suggestions for Authors

This review is well-organized and contains valuable information regarding the significance of the potential of exosomes as therapeutic tools for vascular and neurological disorders. The review covers publications from the last twenty years, which is the most valuable aspect of this review.

Also, the content is well-balanced across different disease models. One minor suggestion I would like to make is that if they could provide references within the figures or tables, it would be more helpful for readers to look up the cited references. For example, in Figure 2, they listed the multifunctional roles of exosomes and their cargos, but there are no references within the figure or in the figure legend. To obtain each piece of information, readers need to go back to the context and find the references. Therefore, providing references in the figures should be recommended.

Author Response

Thank you for the generous comments. We have revised the manuscript as per your advise.

Author's Reply to the Review Report (Reviewer 2)

This review is well-organized and contains valuable information regarding the significance of the potential of exosomes as therapeutic tools for vascular and neurological disorders. The review covers publications from the last twenty years, which is the most valuable aspect of this review.

Also, the content is well-balanced across different disease models. One minor suggestion I would like to make is that if they could provide references within the figures or tables, it would be more helpful for readers to look up the cited references. For example, in Figure 2, they listed the multifunctional roles of exosomes and their cargos, but there are no references within the figure or in the figure legend. To obtain each piece of information, readers need to go back to the context and find the references. Therefore, providing references in the figures should be recommended.

Response:  Thank you for your appreciation and comment. We appreciate your valuable suggestions. We have carefully reviewed your comments and made the necessary corrections in the edited manuscript. References have been added within the figures to enhance the readability and accessibility of the cited information for readers. During the review process, the tables were found to be unnecessary and thus both the whole table have been removed from the edited manuscript. We believe these improvements will enhance the overall clarity and utility of the manuscript.